# Impact of Digital Inequality on the COVID-19 Pandemic: Evidence from European Union Countries

**Marta Borda** [1] , **Natalia Grishchenko** [2] **and Patrycja Kowalczyk-Rólczyńska** [1,*]

1   Department of Insurance, Wroclaw University of Economics and Business, 53345 Wroclaw, Poland;
    marta.borda@ue.wroc.pl
2   Institute of Social Policy, National Research University Higher School of Economics, 101000 Moscow, Russia;
    natalia.b.grishchenko@gmail.com
*   Correspondence: patrycja.kowalczyk@ue.wroc.pl

**Abstract:** One of the consequences of the COVID-19 pandemic is the relationship between social distancing measures and increased use of the Internet, electronic services, and digital devices. How does digital inequality in the context of social distancing affect the COVID-19 pandemic? In this article, we assessed the impact of existing digital inequality as the cause of the changing number of cases of COVID-19 in the EU. We assessed the relationship between the increase in COVID-19 cases between the first and second waves in 2020 and the presence of digital inequality in Internet use and digital skills across sociodemographic factors: gender, age, education, generation, marital status, and place of residence. We applied the ordinary least squares method to data from the 2019 Eurobarometer survey, which reveals the digital maturity of EU citizens, and from the European Center for Disease Prevention and Control in 2020, which tracks COVID-19 cases. We found that the strongest relationship between the number of COVID-19 cases and digital inequality is related to Internet use rather than digital skills. The digital divide by age, between generations, and the geographic digital divide in Internet use show a strong positive relationship with the changing incidence of COVID-19 cases. The gender digital gap shows a negative relationship for both Internet use and digital skills, indicating the social role of women in households in the pandemic, caring for children and the elderly. A negative relation was also found in digital inequality by marital status for digital skills, which reflects preferences regarding living alone during the pandemic. These findings prove the importance of universal access to the Internet for older people and those living in rural areas. The results can contribute to policies aimed at reducing digital inequalities in the face of the ongoing COVID-19 pandemic.

**Keywords:** digital inequality; COVID-19 pandemic; European Union; sociodemographic characteristics; Internet use; digital skills; social distancing



## 1. Introduction

The World Health Organization declared a public health emergency of international concern on 30 January 2020, and a pandemic on 11 March 2020, in response to the worldwide spread of COVID-19. This coincided with the first wave of the pandemic that occurred in the spring of 2020; the second wave came in the fall of 2020. The increase in incidence between these two waves of COVID-19 from weeks 20 to 45 of 2020, according to the data of the European Center for Disease Prevention and Control, shows significant differentiation in European countries from 171 cases per 100,000 population in Ireland, 178 cases in Finland, 244 in Estonia, 4971 in Slovakia, 4801 in Czechia, and 3251 in Bulgaria, respectively [1]. Taking into account the territorial proximity of the EU countries and the general policy of combating the pandemic, this presents an example that can be used for a cross-country assessment of the factors affecting the increases and decreases in the incidence of COVID-19.

In EU countries, both general and additional measures were introduced in 2020 to combat COVID-19, of which the main ones were related to social distancing. These

measures included lockdowns, bans on mass public gatherings, and the closure of schools, entertainment activities, and so on. Since 2020, some non-pharmaceutical interventions have been undertaken by 30 EU and European Economic Area countries, among which the largest category of measures relates to physical distancing, including 'those aimed at reducing contact between individuals: stay-at-home orders and recommendations, private gathering and social circle recommendations, restrictions on public gatherings, and closures of public spaces (e.g. businesses, shops, entertainment venues), closures of public transport, and closures of educational institution (such as day-care centers, primary and secondary schools, universities and higher educational institutions)' [2].

With the introduction of lockdowns from the spring of 2020, the use of the Internet, online communication, remote employment and education, and electronic social services increased, as expected. In 2020, 87% of people aged 16–74 in the EU reported that they had used the Internet during the previous three months. To respect strong social distancing measures in force from March 2020 in most EU countries, one option for staying connected was via telephone or video calls over the Internet. Across all Internet activities, telephone and video calls recorded the largest increase (52% compared with 2019) [3].

In this regard, the effects of social distancing are closely related to digital opportunities, Internet accessibility, digital skills, and the use of digital devices. However, assessing the impact of the digital disparities as one of the predictors of the growth and control of COVID-19 cases has not been sufficiently studied. In this article, we aimed to study the influence of the digital inequality on COVID-19 cases under social distancing during the pandemic. This knowledge will provide an understanding of the required measures to reduce the digital inequality to access online technologies in healthcare, education, employment, and other social areas in a digital society in cases such as a pandemic or crisis.

The article is organized as follows. In the literature review, we provide an overview of recent studies on digital inequalities and social distancing measures during the COVID-19 pandemic. In the section on the method and data, we present the approach and data collection that we use in the study. The next section includes the results of the evaluation carried out. In the conclusion, the findings of the study are discussed.

## 2. Literature Review

Studies on digital inequalities show both its overcoming and its resilience before the COVID-19 pandemic [4–6]. Online employment, education, communication, and entertainment increased because of the social distancing measures during the pandemic, leading to demand for use of the Internet and the application of digital competencies. Recent studies argue that this demand led to the widening of digital inequality during the COVID-19 pandemic.

An assessment of the digital divide by various sociodemographic profiles, personal characteristics, Internet use, and outcomes based on a qualitative survey and quantitative evaluation showed that gender, age, personality, health, literacy, education, economic and social resources, Internet attitude, material access, Internet access, and Internet skills remain important factors in obtaining Internet outcomes in the age of the COVID-19 pandemic [7]. In a representative sample of respondents in the Netherlands studying communication during the pandemic, several groups of people were identified as vulnerable, such as those who are elderly, less educated, or have physical health problems, low literacy levels, or low levels of Internet skills. 'Generally, people who are already relatively advantaged are more likely to use the information and communication opportunities provided by the internet to their benefit in a health pandemic, while less advantaged individuals are less likely to benefit. Therefore, the COVID-19 crisis is also enforcing existing inequalities' [7].

A study of changes in digital communication to reach out to friends and family which took place in the first weeks after lockdown measures were introduced in various parts of the US shows that such characteristics as age, gender, living alone, concerns about Internet access, and Internet skills relate to changes in social contact during the pandemic [8]. Despite an observed vast increase in digital communication, with 46% of respondents

increasing their use of it, 9% of respondents decreased their use of digital communication. Results by socio-demographic characteristics show that the oldest quartile of the sample was more likely to have reduced digital communication than other age groups. Women, those living alone, and those worried about their Internet access were more likely to show increased digital communication. While Internet skills did not cause a significant difference in people's increased use of digital communication, more of the least skilled people decreased their digital communication during the pandemic [8]. Another study showed that people with privileged socioeconomic statuses were more likely increase their Internet skills and online experiences and less likely to decrease their digital communication during the pandemic. The findings illustrate how digital inequalities can put already disadvantaged groups at greater risk of diminished social contact during a public health crisis [9]. The authors emphasized that the global public health crisis may spread unequally among citizens and may continue to shape inequalities even after the pandemic ends.

The digital inequality in terms of age is most pronounced, affecting older people, who are most vulnerable during a pandemic. The study found that their existing level of loneliness and a lack of access to social technologies and the skills and experience to use them effectively were among the challenges faced by older adults when using digital media for social connection during the pandemic. Once online, seniors face the additional challenge of becoming targets of misinformation and fraud in the context of COVID-19 [10]. The results of qualitative in-depth interviews conducted with 30 professionals of at least 60 years of age living in a metropolis in eastern India showed that when participants feel isolated, they change their traditional norms of face-to-face social interaction and rely upon their screens and keyboards to continue everyday interactions, work from home, access information, and use 'essential services'. The study highlighted the blocks to active usage of ICT, namely attitudinal barriers, prior negative experience, concerns over cyber security, complicated technical instructions, and lack of a supportive learning environment [11].

Infrastructure and territorial digital divides persist even in developed countries, affecting connectivity during the COVID-19 pandemic. A recent study explained why advocating for an accessible Internet for all should be a public health and health promotion priority. Broadband Internet access has become a basic need in an interconnected society, linking people to vital resources, such as work, education, health care, food, and information. The study also cited data showing that about a quarter of US adults do not have access to broadband Internet. While the digital disparity in the US is not new, the COVID-19 pandemic has increased societal dependence on the Internet and widened the digital divide. People's reliance on the Internet for access to everything from work and online school to telemedicine and food leads to the fact that, from a social point of view, broadband Internet should be seen as a basic need, not a privilege [12]. A study examining areas in Poland that are particularly vulnerable to digital deprivation due to infrastructure deficiencies during the pandemic indicated that 4% of Poles remain without Internet access, and the access of an additional 10% is too slow to allow effective remote work or learning [13].

Significant inequalities were observed in a study of the availability of online home delivery services in Sweden using the example of providers from the pharmaceutical and food sectors and transport operators delivering parcels [14]. Three groups were studied: the vulnerable population, those marginalized from online home delivery service, and those at risk of contagious disease in physical stores. The results of this study showed that during the COVID-19 pandemic, there are vulnerable members of the population who are challengingly marginalized and at risk of COVID-19 due to the limited accessibility of online home delivery services, the high incidence of COVID-19 in their area, which makes physical visits to a store a risky activity, and their high vulnerability (e.g., high share of individuals older than 65).

Other recent studies looking at digital imbalances during the pandemic highlight the importance of information literacy, data privacy, the use of digital tracing [15–17], misinformation, the proliferation of misleading information, and appropriate regulation [18–20], while the new adoption of surveillance by governments raises concerns about privacy and

data protection [21,22]. The barriers to the use of the technologies mentioned above were mainly due to socioeconomic, educational, and digital inequalities [20]. A study on the impact of Internet information on COVID-19 vaccination rates across US states showed that Internet information might be acting as a largely unheralded enabler. Two cross-state datasets obtained using the Google search engine (one search focused on information on vaccination availability and scheduling in each state, while the other involved information on vaccine reliability and its side effects) show that the greater availability of relevant information on the Internet increased vaccine administration rates, and this was true for both types of Internet searches, resulting in an affirmative answer to the questions above. In contrast, the diffusion of Internet access and the digital divide across states did not have a significant impact on vaccination rates [23]. Using the example of the transformation of scientific communication during the pandemic with the transition from an institutional model to a network model, this can be predicted and taken into account both for other areas and for social groups [24].

A comprehensive list of information technology solutions to combat the COVID-19 pandemic was reviewed in a recent study, including machine learning of real-time information from the computed tomography images from medical records of laboratory-confirmed hospitalized cases [25]. These obviously require both Internet access and digital skills. The important role digital technology plays in mitigating economic disruption due to the pandemic has been proved through assessment of the role of digitization at the time of SARS in 2003. The results are robust in pointing out that those countries with better broadband connectivity were able to mitigate some of the economic losses incurred by the pandemic [26].

A critical issue during the pandemic is the effectiveness of the social distancing policies adopted by states. Recent studies underline the importance of social distancing measures as a tool to control COVID-19 cases. Simulation of the spread and control of COVID-19, tracking the different settings of person-to-person contact (e.g., household, school, workplace), found that there are often long delays between when strong social distancing policies are adopted and when cases and hospitalizations begin to decline [27]. Another study illustrates the potential danger of exponential spread in the absence of social distancing. 'Adoption of government-imposed social distancing measures reduced the daily growth rate of confirmed COVID-19 cases by 5.4 percentage points after one to five days, 6.8 percentage points after six to ten days, 8.2 percentage points after eleven to fifteen days, and 9.1 percentage points after sixteen to twenty days in USA' [28]. The effectiveness of the social distancing measures in preventing the spread of COVID-19 varies by country. One study found that it took one to four weeks from the highest level of promulgation of social distancing measures until the daily confirmed cases showed signs of decreasing for the example of 10 highly infected countries: the US, Spain, Italy, the UK, France, Germany, Russia, Turkey, Iran, and China. The differences in effectiveness are due to the difference in the levels of promulgated social distancing measures and the context of each country [29]. Thus, a critical issue during the novel coronavirus pandemic is the effectiveness of social distancing policies adopted by states, which are dependent on the digital equality of the country.

However, the relationship between the increase in COVID-19 cases and the presence of digital inequalities has not been previously studied. In this article, we examined this relationship considering the previous studies. We evaluated how the spread of COVID-19 may relate to the existing digital inequalities in the EU.

We (1) applied dimensions of the digital inequality, such as Internet use and digital skills, initially using the previous background, (2) determined the relationship between dimensions of the digital inequality and change in COVID-19 cases, taking into account the social and demographic characteristics of citizens, and (3) compared the effects of Internet use and digital skills as key indicators of the digital inequality in the context of COVID-19 pandemic in the EU.

We formulated the following two research questions:

R1. Is there a relationship between increases in COVID-19 cases and digital inequality in Internet use and digital skills related to the social and demographic characteristics of citizens?

R2. What is the relationship between the growth of COVID-19 cases and digital inequality depending on the kinds of digital inequalities?

## 3. Method and Data

We applied ordinary least squares (OLS) estimation to evaluate the relationship between the increase of COVID-19 cases and digital disparities among citizens in the EU. The dependent variable is the increase in cases of COVID-19 between the first and the second waves in 2020. We used the difference between the incidence rates between weeks 20 and 45 in 2020 using the EU example. These data are from the national weekly reports of the European Centre for Disease Prevention and Control [1]. Data refer to the weekly national 14-day notification rate of new COVID-19 cases per 100,000 people by week and country.

The independent variables (predictors) were gender (man, woman), age (25–34; 65–74), generation (1946–1964 'BB'; after 1980 'millennials'), education (end of 15–; end of 20+), marital status (married; single), and place of residence (large town; rural village). Data for the explanatory variables are from the 2019 Eurobarometer survey, which presents EU citizens' opinions on the impact of digitization and automation on daily life [30]. These variables are the self-evaluated use of the Internet and digital skills. Among others, two questions were asked, the answers to which we used in this study. The first question was (1) "how often do you use the Internet?" with an answer: every day. The second question was (2) "to what extent do you agree or disagree with the following statement regarding your skills in the use of digital technologies? You consider yourself to be sufficiently skilled in the use of digital technologies in your daily life", with an answer: totally agree. Descriptive statistics are displayed in Table 1, including: digital inequality in Internet use and digital skills as differences between advantaged (reference group, ref.) and less advantaged social groups by socio-demographic characteristics, for example, between men and women, age 25–34 and 65–74, and so on.

**Table 1.** Descriptive statistics.

| Variable | Mean | Std. Dev. | Min. | Max. |
|---|---|---|---|---|
| Growth in COVID-19 cases per 100,000 population | 1432 | 1392 | 171 | 4971 |
| Internet use, percentage of population | | | | |
| Men | 79 | 9 | 60 | 98 |
| Women | 76 | 9 | 61 | 96 |
| Age (24–35) | 97 | 3 | 91 | 100 |
| Age (65–74) | 48 | 21 | 17 | 93 |
| Generation (1946–1964 'BB') | 60 | 18 | 28 | 96 |
| Generation (After 1980 'millennials') | 97 | 2 | 92 | 100 |
| Education (end of) 15– | 38 | 18 | 14 | 90 |
| Education (end of) 20+ | 91 | 5 | 81 | 99 |
| Married | 77 | 10 | 58 | 98 |
| Single | 90 | 4 | 82 | 96 |
| Rural village | 71 | 12 | 49 | 96 |
| Large town | 84 | 8 | 68 | 98 |

**Table 1.** *Cont.*

| Variable | Mean | Std. Dev. | Min. | Max. |
|---|---|---|---|---|
| Digital skills, percentage of population | | | | |
| Men | 34 | 12 | 17 | 63 |
| Women | 27 | 11 | 11 | 53 |
| Age (24–35) | 47 | 17 | 20 | 84 |
| Age (65–74) | 13 | 10 | 3 | 38 |
| Generation (1946–1964 'BB') | 17 | 11 | 4 | 44 |
| Generation (After 1980 'millennials') | 47 | 16 | 25 | 76 |
| Education (end of) 15– | 12 | 9 | 2 | 37 |
| Education (end of) 20+ | 38 | 10 | 22 | 57 |
| Married | 27 | 12 | 10 | 57 |
| Single | 41 | 11 | 24 | 60 |
| Rural village | 26 | 12 | 7 | 53 |
| Large town | 34 | 13 | 16 | 61 |
| Digital inequality in Internet use | | | | |
| Gender (ref. men) | 4 | 4 | −5 | 12 |
| Age (ref. 25–34) | 50 | 20 | 7 | 77 |
| Generation (Ref. 1980 'millennials') | 37 | 16 | 4 | 65 |
| Education (ref. education (end of) 20+) | 52 | 16 | 9 | 80 |
| Marital status (ref. married) | 1 | 11 | −18 | 25 |
| Territorial (ref. large town) | 13 | 8 | 1 | 29 |
| Digital inequalities in digital skills | | | | |
| Gender (ref. men) | 7 | 3 | 2 | 17 |
| Age (ref. 25–34) | 34 | 11 | 11 | 54 |
| Generation (ref. 1980 'millennials') | 30 | 9 | 12 | 46 |
| Education (ref. education (end of) 20+) | 26 | 8 | 12 | 51 |
| Marital status (ref. married) | 4 | 9 | −12 | 24 |
| Territorial (ref. large town) | 8 | 7 | −3 | 22 |

We applied simple single regression with each predictor. Linear regression attempts to model the relationship between two variables by fitting a linear equation to the observed data. To take into account the heteroscedasticity, a robust variance estimate for standard errors was used. The confidence interval was 0.05 and the number of observations was 28 countries.

## 4. Results

The OLS results which were obtained for the digital divide in Internet use and digital skills showed a more statistically significant relationship in the former case. The relationship between digital inequality and the increase in the number of cases of COVID-19 was associated with four types of digital inequality in the case of Internet use (by gender, age, generation, and place of residence) and with only two types of digital inequality in digital skills (by gender and marital status) (Table 2).

**Table 2.** OLS results.

| Relationship between the Increase in COVID-19 Cases and Digital Inequality | Coef. | R.Std.Err. | t | P > t | R-Squared | Root MSE | Prob > F |
|---|---|---|---|---|---|---|---|
| Internet Use | | | | | | | |
| Gender (ref. men) | −109.47 | 51.50 | −2.13 | 0.043 | 0.085 | 1357.0 | 0.0432 |
| Aged (ref. 25–34) | 35.60 | 9.11 | 3.91 | 0.001 | 0.248 | 1229.7 | 0.0006 |
| Generation (ref. 1980 'millennials') | 41.62 | 10.52 | 3.96 | 0.001 | 0.229 | 1245.7 | 0.0005 |
| Education (ref. education (end of) 20+) | 16.40 | 15.20 | 1.08 | 0.291 | 0.035 | 1393.6 | 0.2907 |
| Marital status (ref. married) | −26.60 | 17.13 | −1.55 | 0.133 | 0.047 | 1385.1 | 0.1326 |
| Territorial (ref. large town) | 73.27 | 22.62 | 3.24 | 0.003 | 0.157 | 1302.4 | 0.0033 |
| Digital skills | | | | | | | |
| Gender (ref. men) | −130.11 | 45.94 | −2.83 | 0.009 | 0.093 | 1350.8 | 0.0088 |
| Aged (ref. 25–34) | −50.94 | 26.94 | −1.89 | 0.070 | 0.156 | 0.1560 | 0.0699 |
| Generation (ref. 1980 'millennials') | −50.32 | 31.33 | −1.61 | 0.120 | 0.106 | 1341.4 | 0.1203 |
| Education (ref. education (end of) 20+) | −46.95 | 26.22 | −1.79 | 0.085 | 0.072 | 1366.5 | 0.0850 |
| Marital status (ref. married) | −53.27 | 26.22 | −2.03 | 0.053 | 0.113 | 1336.0 | 0.0525 |
| Territorial (ref. large town) | −19.33 | 30.16 | −0.64 | 0.527 | 0.008 | 1412.7 | 0.5271 |

The results of OLS regression showed a positive and significant relationship between Internet use and increases in COVID-19 cases in the EU. The strongest relationship existed between digital inequality in Internet use between urban and rural areas and the incidence rate of COVID-19, and there was also a significant positive relationship between digital inequality by age and generation, respectively. In contrast, digital inequality in Internet use by gender showed a negative relationship with increases in the number of COVID-19 cases in the EU countries, that is, where men use the Internet more than women. The same tendency was present regarding the gender digital divide in digital skills. Such a negative relationship between digital inequality in digital skills by marital status and the spread of COVID-19 supports the argument that the more digital skills single people have, the less the pandemic spreads.

To assess these relationships between the digital divide and the spread of the pandemic across the EU, we plot the distributions in graphs by country, showing the predominant relationship between Internet use (Figures 1–4), digital skills (Figures 5 and 6), and COVID-19 cases.

For digital inequalities in Internet use and digital skills identified by socio-demographic characteristics in data from all EU countries and, more importantly, in incidence data of COVID-19 that do not have an exact type of probability distribution, the OLS results were unbiased, noncolinear, and statistically significant for the study's research questions. However, for the same reason, since we used all data, including those with a random distribution of the probability of COVID-19 in different countries, the regression estimates had outliers, which should be considered for prediction. In general, the approach used may be effective taking into account the outliers and limitations of the linear model.

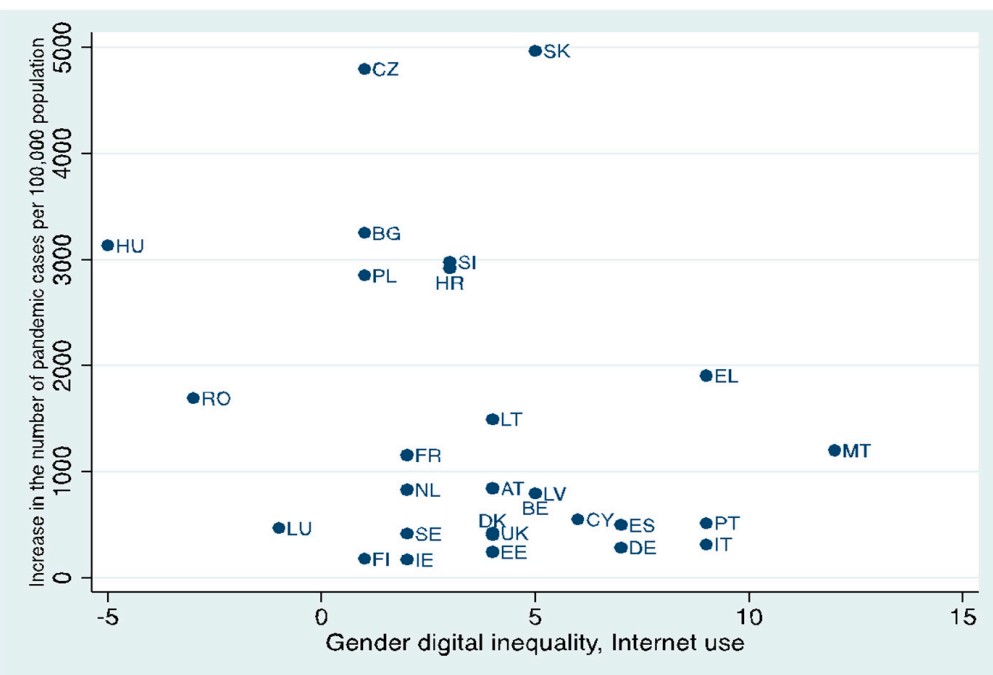

**Figure 1.** The change in COVID-19 cases and gender digital inequality in Internet use, EU, 2020.

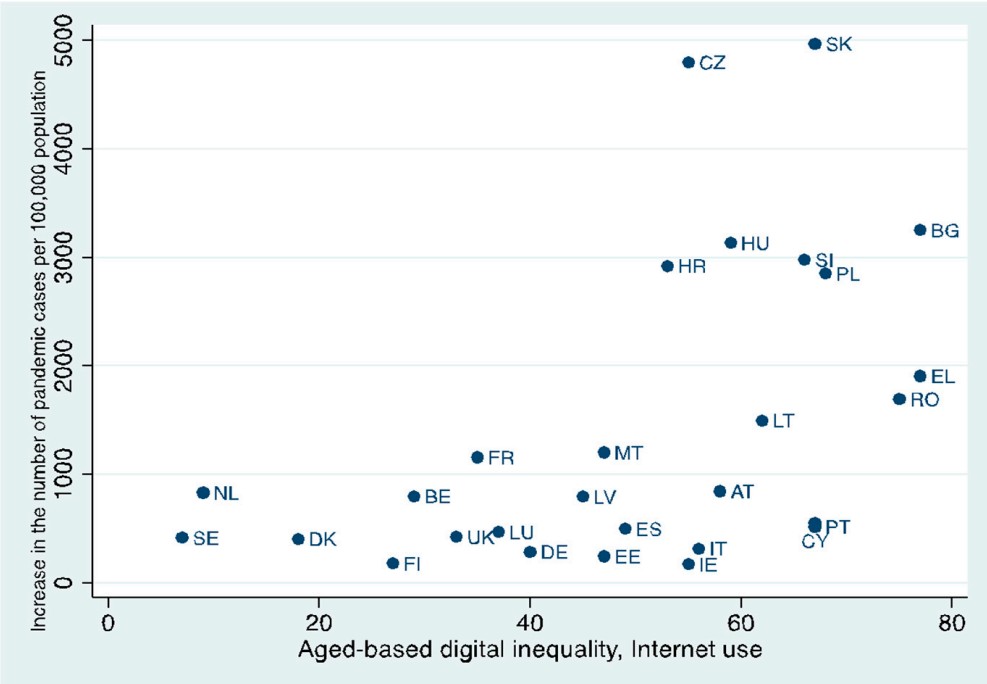

**Figure 2.** The change in COVID-19 cases and age-based digital inequality in Internet use, EU, 2020.

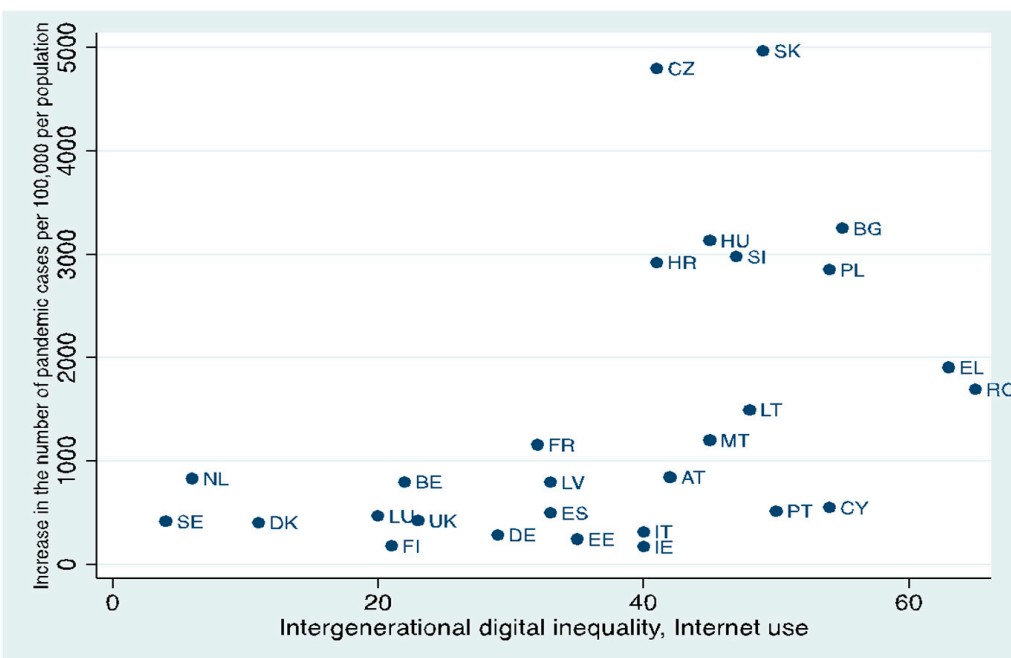

**Figure 3.** The change in COVID-19 cases and intergenerational digital inequality in Internet use, EU, 2020.

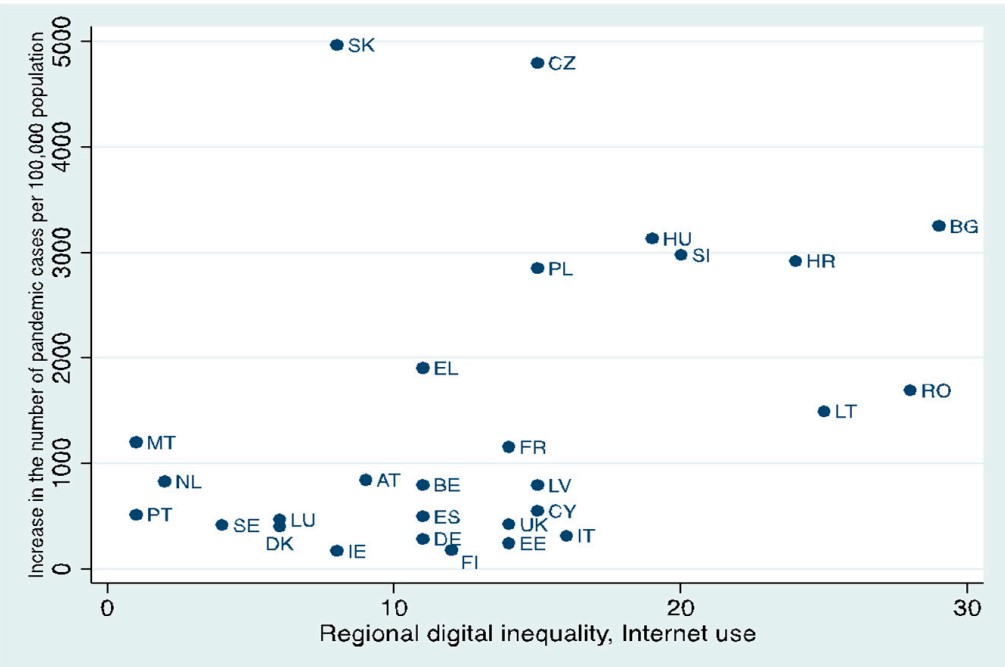

**Figure 4.** The change in COVID-19 cases and territorial digital inequality in Internet use, EU, 2020.

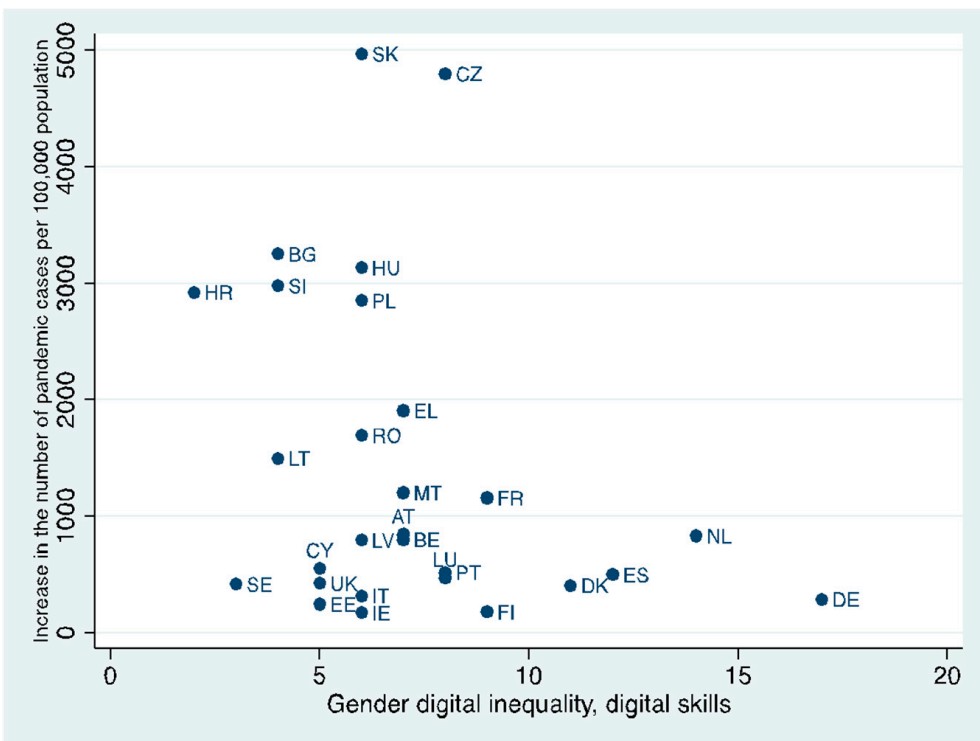

**Figure 5.** The change in COVID-19 cases and gender digital inequality in digital skills, EU, 2020.

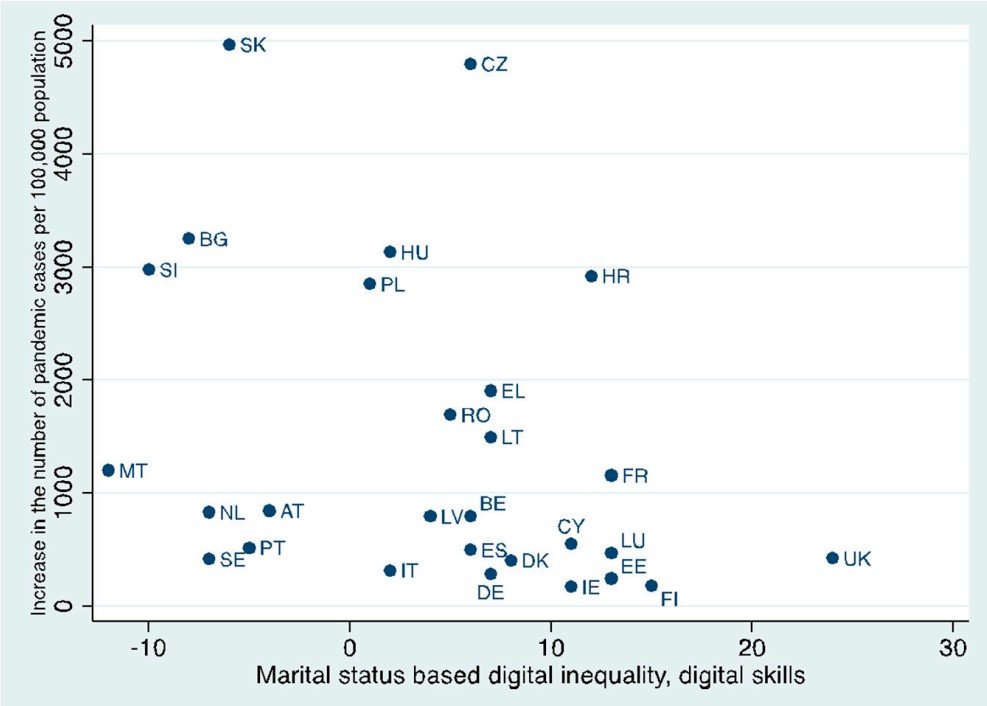

**Figure 6.** The change in COVID-19 cases and marital status digital inequality in digital skills, EU, 2020.

## 5. Discussion and Conclusions

In this article, we assessed the empirical implications of digital inequality in Internet use and digital skills, which may contribute to the number of COVID-19 cases in Europe. We found that digital inequalities by age, generation, place of residence, and gender in Internet use had relationships with the spread of COVID-19 between the first and second waves in

2020. We also found a similar relationship between the divide in digital skills of European citizens by gender and marital status. We received empirically substantiated answers to the research questions posed in the study. These findings suggested that the presence of the digital divide has a significant impact on the spread of COVID-19, which is associated with the ongoing social distancing measures and online forms of work, study, communication, social services, and so on. Our results also allowed us to identify vulnerable groups of people in terms of the digital divide in the context of the pandemic and social distancing, which partially coincide with other recent studies.

We found that the age-related divide in Internet use remains the most important in the control of COVID-19 cases. Older people are the most vulnerable group, with limitations in using information and communication technologies and applications [7–11]. While digital skills are considered important for older people, we found an association only between inequality in Internet use and the incidence of COVID-19. The availability of the Internet is thus a more relevant factor in the control of the pandemic for the elderly. Given that our estimates include an increase in cases between the first and second waves during 2020, that is, in the context of lockdowns in the EU countries, it was the physical availability of the Internet that acted as a more significant factor. As the COVID-19 pandemic and social distancing persist, the risks of digital deprivation remain for older people, and targeted support measures are needed to avoid this. In addition, it is important for the elderly to have access to reliable information from healthcare systems during the pandemic.

The impact of territorial digital inequality in Internet use on the spread of COVID-19 in the EU reflects the presence of challenges regarding the accessibility of the Internet in remote areas [12,13]. The presence of regional digital differences has already had a negative effect on the increase in the number of cases of COVID-19 between the first and second waves of the pandemic in 2020. However, making the Internet universally accessible is also a current challenge in the context of continued social distancing and future lockdowns.

We found the opposite effect of the impact of digital inequality in Internet use and digital skills by gender on changes in the incidence of COVID-19 cases. We suggest that the dominance of men in Internet use and digital skills has an impact on reducing the number of COVID-19 cases due to the important social role that women often play in the context of the public health crisis and social disparity in housekeeping, childcare, and caring for elderly relatives. A recent study also found that men are more likely to use the Internet for COVID-19-related communication [7].

Unlike previous studies, we found no support for the relationship between the digital divide in Internet use and digital skills by educational attainment of European citizens and changes in COVID-19 [7–9]. One of the reasons for this is the sample of respondents, which includes the population over 25 years old and which was not associated with the increase in distance learning in schools and colleges between the first and second waves in 2020. We assume that another reason for this may be a change in behavior during the crisis, with the need to receive more information and to communicate, regardless of the level of education, which also has almost no effect on the availability of modern applications and the use of devices [7].

Overall, we assume that the digital divide has had a significant impact on the prevalence of COVID-19 cases in the period under review. The impact of the digital divide on the control of the pandemic, it can be assumed, will not decrease, given the continuing social distancing and the expansion of online formats of work, study, and communication. Reducing the digital divide, above all by providing universal access to the Internet, could help governments and countries to reduce risks in such crises in the future.

## 6. Limitations and Suggestions for Future Research

The aim of this study was to provide an initial picture of the relationship between the digital inequality and the COVID-19 pandemic. Our study used available data for a limited time of 2019 for the digital divide and 2020 for COVID-19, which constrain the analysis and the method applied. For future research, using broader data with additional dimensions of

the digital inequality and the socioeconomic and demographic characteristics of citizens will provide improved estimates and understanding of the impact of the digital divide on the pandemic and crisis.

Since the EU was an example of the available data, this may also give particular results. Moreover, we used averages, excluding country differences in the EU, which provides average estimates.

Other limitations are related to the contextual form of the study and the method used. Taking into account the context of the pandemic, other health care indicators could be applied to compare the influence of the digital divide on COVID-19 cases. The OLS method used in the study has its limitations associated with the assumption of a linear model for all research subjects and the exclusion of a possible magnitude in the values of the variables under study.

**Author Contributions:** Conceptualization, M.B., N.G. and P.K.-R.; Formal analysis, N.G.; Methodology, N.G.; Writing—original draft, M.B., N.G. and P.K.-R.; Writing—review & editing, M.B., N.G. and P.K.-R. All authors have read and agreed to the published version of the manuscript.

**Funding:** The project is financed by the Ministry of Science and Higher Education in Poland under the 334 programme "Regional Initiative of Excellence" 2019–2022 project number 015/RID/2018/19 total 335 funding amount 10 721 040,00 PLN.

**Data Availability Statement:** Publicly available datasets were analyzed in this study. These data can be found here: [European Centre for Disease Prevention and Control, https://www.ecdc.europa.eu/en/publications-data/14-day-case-notification-rate-100-000-inhabitants-updated-20-january-2022, accessed on 24 February 2022; Response measures database. European commission, https://covid-statistics.jrc.ec.europa.eu/RMeasures, accessed on 24 February 2022; Attitudes towards the impact of digitisation and automation of daily life. 2019. Eurobarometer survey, https://digital-strategy.ec.europa.eu/en/news/attitudes-towards-impact-digitisation-and-automation-daily-life], accessed on 24 February 2022.

**Conflicts of Interest:** The authors declared no potential conflict of interest with respect to the research, authorship, and/or publication of this article.

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
