# Peer review of "Impact of Digital Inequality on the COVID-19 Pandemic: Evidence from European Union Countries"

_sustainability, doi:10.3390/su14052850_

Round 1

Reviewer 1 Report

The paper is very interesting and addresses an important challenge. The paper is well written, and after some minor corrections, it can be more perfect.
In the title, instead of EU, use the full name.
Please clarify the research aim, objective, and research questions in the introduction.
The limitation section is too brief, while more limitation exists for this type of study.

Author Response

Dear Reviewer,

Thank you very much for your review. We are grateful for your constructive comments, which help to improve the article. We agree with all comments, the changes made (in italics) are below.

In the title, instead of EU, use the full name.

We have changed the title to “Impact of digital inequality on the Covid-19 pandemic: evidence from European Union countries”  p.1

Please clarify the research aim, objective, and research questions in the introduction.

We have made more clear research aim, objective, and research questions in the Introduction and Literature review.

Research aim: In this article, we aim to study the influence of the digital inequality on Covid-19 cases under social distancing during the pandemic. This knowledge will provide an understanding of the required measures to reduce the digital inequality to access online technologies in healthcare, education, employment, and other social areas in a digital society in cases such as a pandemic or crisis.  p.2

Research objective: We (1) apply dimensions of the digital inequality, such as Internet use and digital skills, initially using the previous background, (2) determine the relationship between dimensions of the digital inequality and change in Covid-19 cases, taking into account the social and demographic characteristics of citizens, and (3) compare the effects of Internet use and digital skills as key indicators of the digital inequality in the context of Covid-19 pandemic in the EU.  p. 5

Research questions: R1. Is there a relationship between increases in Covid-19 cases and digital inequality in Internet use and digital skills related to the social and demographic characteristics of citizens?

R2. What is the relationship between the growth of Covid-19 cases and digital inequality depending on the kinds of digital inequalities?   p.5

The limitation section is too brief, while more limitation exists for this type of study.

We have clarified and added suggestions to the section Limitations and suggestions for future research.

For future research, using broader data with additional dimensions of the digital inequality and the socioeconomic and demographic characteristics of citizens will provide improved estimates and understanding of the impact of the digital divide on the pandemic and crisis.

Other limitations are related to the contextual form of the study and the method used. Taking into account the context of the pandemic, other health care indicators could be applied to compare the influence of the digital divide on Covid-19 cases. The OLS method used in the study has its limitations associated with the assumption of a linear model for all research subjects and the exclusion of a possible magnitude in the values of the variables under study.  p.12

Thank you for your consideration of our article.

Kind regards,

Authors

Reviewer 2 Report

The draft is well written and the literature is inclusive and up-to-date. I think that the study makes important contributions to the literature. Congratulations to the authors.

Author Response

Thank you very much for your review.